# From Boltzmann to Zipf through Shannon and Jaynes

**DOI:** 10.3390/e22020179

**Published:** 2020-02-05

**Authors:** Álvaro Corral, Montserrat García del Muro

**Affiliations:** 1Centre de Recerca Matemàtica, Edifici C, Campus Bellaterra, E-08193 Barcelona, Spain; acorral@crm.cat; 2Departament de Matemàtiques, Facultat de Ciències, Universitat Autònoma de Barcelona, E-08193 Barcelona, Spain; 3Barcelona Graduate School of Mathematics, Edifici C, Campus Bellaterra, E-08193 Barcelona, Spain; 4Complexity Science Hub Vienna, Josefstädter Stra*β*e 39, 1080 Vienna, Austria; 5Departament de Física de la Matèria Condensada, Universitat de Barcelona, Martí i Franquès 1, E-08028 Barcelona, Spain; 6IN2UB, Universitat de Barcelona, Martí i Franquès 1, E-08028 Barcelona, Spain

**Keywords:** maximum entropy principle, two-letter interactions, Boltzmann factor, word-frequency distribution, Zipf’s law, quantitative linguistics, power laws

## Abstract

The word-frequency distribution provides the fundamental building blocks that generate discourse in natural language. It is well known, from empirical evidence, that the word-frequency distribution of almost any text is described by Zipf’s law, at least approximately. Following Stephens and Bialek (2010), we interpret the frequency of any word as arising from the interaction potentials between its constituent letters. Indeed, Jaynes’ maximum-entropy principle, with the constrains given by every empirical two-letter marginal distribution, leads to a Boltzmann distribution for word probabilities, with an energy-like function given by the sum of the all-to-all pairwise (two-letter) potentials. The so-called improved iterative-scaling algorithm allows us finding the potentials from the empirical two-letter marginals. We considerably extend Stephens and Bialek’s results, applying this formalism to words with length of up to six letters from the English subset of the recently created Standardized Project Gutenberg Corpus. We find that the model is able to reproduce Zipf’s law, but with some limitations: the general Zipf’s power-law regime is obtained, but the probability of individual words shows considerable scattering. In this way, a pure statistical-physics framework is used to describe the probabilities of words. As a by-product, we find that both the empirical two-letter marginal distributions and the interaction-potential distributions follow well-defined statistical laws.

## 1. Introduction

Zipf’s law is a pattern that emerges in many complex systems composed by individual elements that can be grouped into different classes or types [1]. It has been reported in demography, with citizens linked to the city or village where they live [2]; in sociology, with believers gathering into religions [3]; in economy, with employees hired by companies [4]; and also in ecology [5,6], communications [3,7], cell biology [8], and even music [9,10,11]. In all these cases, the size of the groups in terms of the number of its constituent elements shows extremely large variability, more or less well described in some range of sizes by a power-law distribution with an exponent close to two (for the probability mass function; this turns out to be an exponent close to one for the complementary cumulative distribution).

Of particular interest is Zipf’s law in linguistics [12,13,14,15,16,17], for which individual elements are word tokens (i.e., word occurrences in a text), and classes or groups are the words themselves (word types). In this way, the “size” of a word type is given by the number of tokens of it that appear in the text under study (in other words, the absolute frequency of the word), and thus, the linguistic version of the law states that the frequency of word types can be described by a power-law probability mass function, with an exponent around two. Some variability has been found in the value of the exponent regarding the language [18] or age of the speakers [19], but not the length of the text [20,21] or the precise word definition (i.e., word forms versus lemmas [20]). Let us clarify that (what we call today) Zipf’s law was discovered more than 100 years ago by Estoup [22] and formalized mathematically by the recognized physicist E. Condon in 1928 [23]; the posterior rediscovery and intensive work by Zipf [24] is what made the law so well-known.

There have been many attempts to provide a mechanism for this curious law [25,26,27]. With text generation in mind, we can mention monkey typing, also called intermittent silence [28] (criticized in [29]), the least effort principle [30,31,32], sample-space reduction [33,34], and codification optimization [35]. More general mechanistic models for Zipf’s law are preferential attachment [18,36,37,38], birth-and-death processes [39], variations of Polya urns [40] and random walks on networks [41]. The existence of so-many models and explanations is a clear indication of the controversial origin of the law. Furthermore, there have been also important attempts to explain not only Zipf’s law but any sort of power-law distributions in nature [42,43,44,45].

A different approach is provided by the maximum-entropy principle. In statistical physics it is well known that a closed system in equilibrium with a thermal bath displays fluctuations in its energy but keeping a constant mean energy. As Jaynes showed [46], the maximization of the Shannon entropy [47] with the constrain that the mean energy is fixed yields the Boltzmann factor, which states that the probability of any microstate has to be an exponential function of its energy (note that this does not mean that the distribution of energy is exponential, as the number of microstates as a function of the energy is not necessarily constant).

Therefore, some authors have looked for an analogous of the Boltzmann factor for power laws. For example, one can easily obtain a power law not imposing a constant (arithmetic) mean but a constant geometric mean [48] (assuming also a degeneracy that is constant with respect the energy). Also, fixing both the arithmetic and the geometric mean leads to a power law with an exponential tail [49]. Nevertheless, the physical meaning of these constraints is difficult to justify.

More recently, Peterson et al. [50] have proposed a concrete non-extensive energy function that leads to power-law tails of sizes when maximizing the Shannon entropy. The main idea is that the probability is exponential with the energy, but the energy is logarithmic with size, resulting in an overall power law for sizes [50]. Other authors have found the use of the Shannon entropy inadequate, due to its close connection with exponential distributions, and have generalized the very entropy concept, yielding non-extensive entropies such as the Havrda-Charvát entropies [51], also called Tsallis entropies [52], and the Hanel-Thurner entropies [53,54].

Here we will follow the distinct approach of Stephens and Bialek [55], extending their results. Like Peterson et al. [50], these authors [55] consider the well-known Jaynes’ maximization of the plain Shannon entropy, but in contrast to them [50], no functional form is proposed a priori for the energy. Instead, the constrains are provided by the empirical two-body marginal distributions. The framework is that of word occurrence in texts, and words are considered to be composed by letters that interact all to all, in pairs. In a physical analogy, one could think of a word as a (one-dimensional) molecule, and the constituent letters would be the corresponding atoms. The interaction between atoms (letters) does not only depend on the distance but also on the position (i.e., the interaction between a and b is not the same between positions 1 and 3 than between 2 and 4, and so on; moreover, symmetry is not preserved). Let us remark that this is different from a Markov model [47]; all-to-all interaction is an important distinction. The resulting Boltzmann-like factor allows one to identify, in a natural way, the Lagrange multipliers (obtained in the maximization of entropy under the empirical values of the constrains) with the interaction potentials (with a negative sign).

Stephens and Bialek [55] only considered four-letter English words and performed a visual comparison with the empirical frequencies of words. We will considerably extend their results by analyzing words of any length from 1 to 6 letters in a much larger English corpus, and will undertake a quantitative statistical analysis of the fulfillment of Zipf’s law. In this way, using Shannon and Jaynes’ framework we will obtain a Boltzmann-like factor for the word probabilities that will allow a direct comparison with Zipf’s law. We will pay special attention to the values of the interaction potentials. The main conclusion is that two-body (two-letter) pairwise interactions are able to reproduce a power-law regime for the probabilities of words (which is the hallmark of Zipf’s law), but with considerable scatter of the concrete values of the probabilities.

In the next section, we review the maximum-entropy formalism and its application to pairwise interaction of letters in words, using the useful concept of feature functions. Next, we describe the empirical data we use and the results, including the empirical pairwise marginals (which are the input of the procedure) and the resulting pairwise potentials (which are the output from which the theoretical word distribution is built). The Zipfian character of the theoretical word distribution as well as its correspondence with the empirical distribution is evaluated. In the final section we discuss limitations and extensions of this work.

## 2. Maximum Entropy and Pairwise Interactions

“Information theory provides a constructive criterion for setting up probability distributions on the basis of partial knowledge,” which leads to a special type of statistical inference. This is the key idea of Jaynes’ maximum-entropy principle [46]. The recipe can be summarized as: use that probability distribution which has maximum Shannon entropy subject to whatever is known. In other words, everything should be made as random as possible, but no more [56] (E. G. Altmann has made us notice that Jaynes, being close to be a Bayesian, would not have totally agreed with the identification of entropy with randomness, and would have prefer the use of “ignorance”. Therefore, we could write instead: we should be as ignorant as possible, but no more.).

Let us consider words in texts. Labelling each word type by *j*, with j=1,2,…V, and *V* the size of the vocabulary (the total number of word types), the Shannon entropy is
$$S=-\sum_{j=1}^{V}P_{j}\ln P_{j},$$
where $P_{j}$ is the probability of occurrence of word type *j*. Please note that as we use natural logarithms, the entropy is not measured in bits but in nats, in principle. To maximize the entropy under a series of constrains one uses the method of Lagrange multipliers, where one finds the solution of
(1)$$\frac{\partial L}{\partial P_{j}}=-\ln P_{j}-1-\alpha \frac{\partial }{\partial P_{j}}\left(\mathrm{constrain}1\right)-\beta \frac{\partial }{\partial P_{j}}\left(\mathrm{constrain}2\right)-\cdots =0,$$
for all *j*, with α, β, etc., the Lagrange multipliers associated with constrain 1, constrain 2, etc., and L=S−α×(constrain1)−β×(constrain2)−… the Lagrangian function.

One can see that the maximum-entropy method yields intuitive solutions in very simple cases. For example, if no constrains are provided one obtains the equiprobability case, $P_{j}^{\mu c}=1/V$ (as there is in fact one implicit constrain: normalization; μc stands from microcanonical, in analogy with statistical physics). If there are no other constrains it is clear one cannot escape this “rudimentary” solution. If, instead, one uses all empirical values as constrains, one gets the same one puts, with a solution $P_{j}^{full}=\rho \left(j\right)$, with ρ(j) the empirical probability of occurrence of word *j* (i.e., the relative frequency of *j*). Therefore, the full data is the solution, which is of little practical interest, as this model lacks generalization and does not bring any understanding.

More interestingly, when the mean value of the energy is used as a constrain (as it happens in thermodynamics for closed systems in thermal equilibrium with a bath), the solution is given by the Boltzmann distribution [57],
(2)$$P_{j}^{can}=\frac{e^{-\beta E_{j}}}{Z},$$
with the notation can coming from the analogy with the canonical ensemble, $E_{j}$ referring to the energy of type (or state) *j*, and with $Z=\sum_{j}e^{-\beta E_{j}}$. If one could propose, a priori, an expression for the energy $E_{j}$ of a word type, the word probability would follow immediately; however, that energy would lack interpretation. Let us stress that we are not referring to the physical (acoustic) energy [58,59,60], as arising in speech; in fact, our approach (which is that of Stephens and Bialek) would lead to something analogous to the energy of a word. However, the analogy one finds in this way (through the Boltzmann factor) is so neat that it is not possible to escape identifying that with a sort of “energy”.

### 2.1. Feature Functions and Marginal Probabilities

At this point it becomes useful to introduce the feature functions [61]. Given a feature *i*, the feature function $f_{i}\left(j\right)$ is a function that for each word *j* takes the values
$$f_{i}\left(j\right)=\left\{\begin{array}{cc} 1 & \mathrm{if}\;\mathrm{the}\;\mathrm{word}\;j\;\mathrm{contains}\;\mathrm{feature}\;i\\ 0 & \mathrm{if}\;\mathrm{not} \end{array}\right.$$

For example, let us consider the feature i={ letter c is in position 1}, summarized as i=1c (in this case, this could be called letter function); then $f_{1c}\left(\mathrm{cat}\right)=1$ and $f_{1c}\left(\mathrm{mice}\right)=0$, as c is the first letter in cat but not in mice (let us mention that, for us, capital and lower-case letters are considered the same letter).

Considering *m* features, each one yielding a constrain for its expected value, we have
(3)$$\langle f_{i}\rangle =\sum_{j=1}^{V}P_{j}f_{i}\left(j\right)=\sum_{j\;s.t.\;i\;in\;j}P_{j}=F_{i}$$
for i=1,2,…m, with $F_{i}$ the empirical mean value of feature *i* (fraction of word tokens with feature *i*). Please note that $P_{j}$ and $\langle f_{i}\rangle$ are unknown, whereas $F_{i}$ should not. With these *m* constrains, the method of Lagrange multipliers [Equation (Equation 1)] leads to
$$\frac{\partial L}{\partial P_{j}}=-\ln P_{j}-1+\sum_{i=1}^{m}\lambda_{i}f_{i}\left(j\right)=0,$$
where $\lambda_{i}$ are now the Lagrange multipliers (we have in fact inverted their sign with respect the previous cases, in particular Equations (Equation 1) and (Equation 2), for convenience). The solution is
(4)$$P_{j}=\exp\left(-1+\sum_{i=1}^{m}\lambda_{i}f_{i}\left(j\right)\right)$$
$$=\exp\left(-1+\sum \lambda {}^{\prime }s\;\mathrm{of}\;\mathrm{features}\;\mathrm{of}\;\mathrm{word}\;j\right).$$

In contrast with the previous simplistic models, we are now able to deal with the inner structure of words, as composed by letters, i.e., $j=\{\ell_{1},\ell_{2},\ldots \}$ and $P_{j}=P\left(\ell_{1}\ell_{2}\ldots \right),$ with $\ell_{1}$ the letter at first position of word *j* and so on. If we consider that the features describe the individual letters of a word, for example, for i=1c, then Equation (Equation 3) writes
(5)$$\langle f_{1c}\rangle =\sum_{j=1}^{V}P_{j}f_{1c}\left(j\right)=\sum_{\ell_{2}=a}^{z}\sum_{\ell_{3}=a}^{z}\cdots P\left(c\ell_{2}\ell_{3}\ldots \right)=P_{1}^{I}\left(c\right)=\rho_{1}\left(c\right)$$

(using that only words starting with $\ell_{1}=c$ contribute to the sum); in words, we obtain that the expected value of the feature 1c is the marginal probability $P_{1}^{I}\left(c\right)$ that the first letter in a word is c, which we make equal to its empirical value $\rho_{1}\left(c\right)$ (which is just the number of tokens with letter c in position 1 divided by the total number of tokens). Notice that we do not impose normalization constrain for the $P_{j}$’s, as this is implicit in the marginals.

Coming back to the expression for the probabilities, Equation (Equation 4), we have, for a three-letter example,
$$P^{I}\left(\mathrm{cat}\right)=\exp\left(\lambda_{1c}+\lambda_{2a}+\lambda_{3t}-1\right),$$
the label *I* standing for the fact that the solution is obtained from the constrains of one-letter marginals. Substituting this into the constrain, Equation (Equation 5), we arrive to $\rho_{1}\left(c\right)\propto e^{\lambda_{1c}}$, from where we can take solutions of the form $e^{\lambda_{1c}-1/3}=\rho_{1}\left(c\right)$ and so,
$$P^{I}\left(\mathrm{cat}\right)=\rho_{1}\left(c\right)\rho_{2}\left(a\right)\rho_{3}\left(t\right)$$

(note that other solutions for the $\lambda_{1c}$’s are possible, but they lead to the same $P^{I}$’s; in particular, the origin of each potential is not fixed and one could replace, for instance, $\lambda_{1\ell_{1}}\to \lambda_{1\ell_{1}}+C_{1}$ for all $\ell_{1}$, provided that the other potentials are modified accordingly to yield the same value of the sum).

This model based on univariate (single-letter) marginals is very simple indeed, and closely related to monkey-typing models [28,29], as we obtain that each word is an independent combination of letters, with each letter having its own probability of occurrence (but depending on its position in the word). In other words, one could think of a monkey typing on a keyboard at random, with each letter having a different probability. However, in addition, the probabilities of each letter change depending if the letter is the first of a word, or the second, etc. (the blank is considered to be special letter, which signals the end of a word). Please note that, although the “classical” extension of these simple monkey-typing models is towards Markov models [47], Stephens and Bialek’s approach [55] takes a different direction.

### 2.2. Pairwise Constrains

The approach of Stephens and Bialek uses the generalization of the previous model to two-letter features, which leads to constrains over the two-letter marginals. For instance, if the feature i=12ca denotes that the word has letter c in position 1 and letter a in 2, then, Equation (Equation 3) writes
(6)$$\langle f_{12\mathrm{ca}}\rangle =\sum_{\forall j}P_{j}f_{12\mathrm{ca}}\left(j\right)=\sum_{\ell_{3}=a}^{z}\sum_{\ell_{4}=a}^{z}\cdots P\left(\mathrm{ca}\ell_{3}\ldots \right)=P_{12}^{II}\left(\mathrm{ca}\right)=\rho_{12}\left(\mathrm{ca}\right),$$
with $\rho_{12}\left(\mathrm{ca}\right)$ the two-letter marginal, provided by the empirical data,
$$\rho_{12}\left(\mathrm{ca}\right)=\frac{\mathrm{number}\;\mathrm{of}\;\mathrm{tokens}\;\mathrm{with}\;c\mathrm{in}\;1\mathrm{and}\;a\;\mathrm{in}\;2}{\mathrm{total}\;\mathrm{number}\;\mathrm{of}\;\mathrm{tokens}}.$$

The solution (Equation 4), restricted for the particular example of a three-letter word can be written as
(7)$$P^{II}\left(\mathrm{cat}\right)=\exp\left(\lambda_{12}\left(\mathrm{ca}\right)+\lambda_{13}\left(\mathrm{ct}\right)+\lambda_{23}\left(\mathrm{at}\right)-1\right),$$
using the notation $\lambda_{12\mathrm{ca}}=\lambda_{12}\left(\mathrm{ca}\right)$ for the multipliers, and the label II denoting that we are dealing with theoretical probabilities arising from two-letter features, i.e., two-letter marginals. The same result writes, in general,
(8)$$P^{II}\left(\ell_{1}\ell_{2}\ldots \ell_{K}\right)=\exp\left(-1+\sum_{k=1}^{K-1}\sum_{k^{\prime }=k+1}^{K}\lambda_{kk^{\prime }}\left(\ell_{k}\ell_{k^{\prime }}\right)\right),$$
with *K* the word length (in number of letters). Comparing to Boltzmann distribution, as in Equation (Equation 2), we can identify the argument of the exponential with the energy (in units of $\beta^{-1}$ and with a minus sign) and the Lagrange multiplier for each feature with the pairwise interaction potential between the letters defining the feature (with a minus sign, and with a shift of one unit); for example,
$$-\beta E\left(\mathrm{cat}\right)=\lambda_{12}\left(\mathrm{ca}\right)+\lambda_{13}\left(\mathrm{ct}\right)+\lambda_{23}\left(\mathrm{at}\right)-1,$$
and in general,
$$-\beta E\left(\ell_{1}\ell_{2}\ldots \ell_{K}\right)=-1+\sum_{k=1}^{K-1}\sum_{k^{\prime }=k+1}^{K}\lambda_{kk^{\prime }}\left(\ell_{k}\ell_{k^{\prime }}\right).$$

Therefore, words can be seen as networks of interacting letters (with all-to-all interaction between pairs, and where the position of the two letters matters for the interaction). Please note that three-letter interactions, common in English orthographic rules, are not captured by the pairwise interaction; for example, in positions 3 to 5: believe (rule) versus deceive (exception, due to the c letter). Remarkably, this pairwise approach has been used also for neuronal, biochemical, and genetic networks [55]. A very simplified case of this letter system turns out to be equivalent to an Ising model (or, more properly, a spin-glass model): just consider an alphabet of two letters (a and b) and impose the symmetries (not present in linguistic data, in general) $\lambda_{kk^{\prime }}\left(ab\right)=\lambda_{kk^{\prime }}\left(ba\right)$ and $\lambda_{kk^{\prime }}\left(aa\right)=\lambda_{kk^{\prime }}\left(bb\right)$ (if one wants to get rid of this symmetry in the Ising system one could consider external “magnetic” fields, associated with the one-letter marginals).

Substituting the solution (Equation 4) or (Equation 7) into the constrains (Equation 6), the equations we need to solve would be like
$$P_{12}^{II}\left(\mathrm{ca}\right)=\langle f_{12\mathrm{ca}}\rangle =\sum_{j}f_{12\mathrm{ca}}\left(j\right)\;e^{\;-1+\sum_{i=1}^{m}\lambda_{i}f_{i}\left(j\right)}=$$
(9)$$=e^{\lambda_{12}\left(\mathrm{ca}\right)}\sum_{\ell_{3}=a}^{z}e^{\lambda_{13}\left(c\ell_{3}\right)+\lambda_{23}\left(a\ell_{3}\right)-1}=\rho_{12}\left(\mathrm{ca}\right),$$
if we restricted to three-letter words.

For computational limitations, we will only treat words comprising from 1 to 6 letters. As the numerical algorithm we will use requires that the number of letters is constant (see the Appendix A), we will consider that words shorter than length 6 are six-letter words whose last positions are filled with blanks; for example, cat = cat□□□, where the symbol □ denotes a blank. In this way, instead of the usual 26 letters in English we deal with 27 (the last term in the sums of some of the previous equations should be □, instead of z). This yields 6×5/2=15 interaction potentials (15 features) for each word, and a total of $15\times 27^{2}=10,935$ unknown values of the interaction potential (i.e., Lagrange multipliers with minus sign) corresponding to 10,935 equations (one for each value of the two-letter marginals). In contrast, note that there are about $27^{6}=387,420,489$ possible words of length between 1 and 6 (the figures turn out to be a bit smaller if one recalls that blanks can only be at the end of the word, in fact, $26+\ldots +26^{6}=321,272,406$). In more generality, the 10,935 equations to solve are like
(10)$$e^{\lambda_{12}\left(\mathrm{ca}\right)}\sum_{\ell_{3}\ldots \ell_{6}}e^{\lambda_{13}\left(c\ell_{3}\right)+\cdots +\lambda_{16}\left(c\ell_{6}\right)+\lambda_{23}\left(a\ell_{3}\right)+\cdots +\lambda_{26}\left(a\ell_{6}\right)+\lambda_{34}\left(\ell_{3}\ell_{4}\right)+\ldots \cdots +\lambda_{56}\left(\ell_{5}\ell_{6}\right)-1}=\rho_{12}\left(\mathrm{ca}\right),$$
where the solution is not straightforward anymore, and has to be found numerically. Therefore, we deal with a constrained optimization problem, for which the Appendix A provides complete information. Here we just mention that the so-called improved iterative-scaling method [61,62] consist of the successive application of transformations as
$$\lambda_{12}\left(\mathrm{ca}\right)\to \lambda_{12}\left(\mathrm{ca}\right)+\frac{1}{15}\ln\frac{\rho_{12}\left(\mathrm{ca}\right)}{P_{12}^{II}\left(\mathrm{ca}\right)},$$
see Equation (Equation 14) in the Appendix A, with $P_{12}^{II}\left(\mathrm{ca}\right)$ calculated from the maximum-entropy solution, Equation (Equation 9). Please note that, as in the case of univariate marginals, the potentials are undetermined under a shift, i.e., $\lambda_{12}\left(\ell_{1}\ell_{2}\right)\to \lambda_{12}\left(\ell_{1}\ell_{2}\right)+C_{12}$, as long as the other potentials are correspondingly shifted to give the same value for the sum.

## 3. Data and Results

### 3.1. Data

As a corpus, we use all English books in the recently presented Standardized Project Gutenberg Corpus [63]. This comprises more than 40,000 books in English, with a total number of tokens 2,016,391,406 and a vocabulary size V=2,268,043. The entropy of the corresponding word-probability distribution is S=10.27 bits. To avoid spurious words (misspellings, etc.), and also for computational limitations, we disregard word types with absolute frequency smaller than 10,000; the corresponding relative frequencies are below $5\times 10^{-6}$. Also, word types (unigrams) containing characters different than the plain 26 letters from a to z are disregarded (note that we do not distinguish between capital and lower-case letters). Finally, we remove also Roman numerals (these are not words for our purposes, as they are not formed by interaction between letters). This reduces the number of tokens to 1,881,679,476 and *V* to 11,042, and so the entropy becomes S=9.45 bits. Finally, the subset of words with length smaller or equal to 6 yields 1,597,358,419 tokens, V=5081 and S=8.35 bits. We will see that these sub-corpora fulfill Zipf’s law, but each one with a slightly different power-law exponent. Please note that the fact of disregarding relative frequencies below $5\times 10^{-6}$ does not influence the fulfillment of Zipf’s law, as Zipf’s law is a high-frequency phenomenon (see Table 1).

### 3.2. Marginal Distributions

Figure 1 displays the empirical two-letter marginal probabilities (obtained from the 6-or-less-letter sub-corpus just described), which constitute the target of the optimization procedure. There are a total of 5092 non-zero values of the marginals. Notice that, although the two-letter marginals are bivariate probabilities (for example, $\rho_{12}\left(\ell_{1}\ell_{2}\right)$, see also Figure 1a in [55]), Zipf’s representation allows one to display them as univariated. This is achieved by defining a rank variable, assigning rank r=1 to the type with the highest empirical frequency ρ (i.e., the most common type), r=2 to the second most common type, and so on (Figure 1(left)). This is called the rank-frequency representation (or, sometimes, distribution of ranks), and constitutes a sort of projection of a bivariate (in this case, or multivariate, in general) distribution into a univariate one; for example, $\rho_{12}\left(\ell_{1}\ell_{2}\right)$, instead of being represented in terms of the random variables $\ell_{1}$ and $\ell_{2}$, is considered a univariate function or the rank, $\rho_{12}\left(r\right)$.

Then, Zipf’s law can be formulated as a power-law relation between ρ and *r*,
(11)$$\rho \propto \frac{1}{r^{1/\gamma }}$$
for some range of ranks (typpically the lowest ones, i.e., the highest frequencies), with the exponent $\gamma^{-1}$ taking values close to one (the symbol ∝ denotes proportionality). When we calculate and report entropies we use always the rank-frequency representation.

An approximated alternative representation [17,64,65], also used by Zipf [14,24], considers the empirical frequency ρ as a random variable, whose distribution is computed. In terms of the complementary cumulative distribution, G(ρ), Zipf’s law can be written as
(12)$$G\left(\rho \right)\propto \frac{1}{\rho^{\gamma }},$$
which in terms of the probability density or probability mass function of ρ leads to
(13)$$g\left(\rho \right)\propto \frac{1}{\rho^{\gamma +1}},$$
asymptotically, for large ρ (Figure 1(right)). Both G(ρ) and g(ρ) constitute a representation in terms of the distribution of frequencies. For more subtle arguments relating ρ(r), G(ρ), and g(ρ), see [17,64,65].

We can test the applicability of Zipf’s law to our two-letter marginals, in order to evaluate how surprising or unsurprising is the emergence from them of Zipf’s law in the word distribution. Remember that, in the case of marginal distributions, types are pairs of letters. Figure 1(left) shows that, despite the number of data in the marginals is relatively low (a few hundred as shown in Table 1, with a theoretical maximum equal to $26^{2}=676$), the marginal frequencies appear as broadly distributed, varying along 4 orders of magnitude (with the frequency ρ in the range from $10^{-5}$ to $10^{-1}$). Although the double logarithmic plots do not correspond to straight lines, the high-frequency (low-rank) part of each distribution can be fitted to a power law, for several orders of magnitude ranging from 0.5 to 2 and an exponent γ typically between 1 and 2, as it can be seen in Table 1. Thus, the two-letter marginal distributions display a certain Zipfian character (at least considering words of length not larger than 6, in letters), with a short power-law range, in general, and with a somewhat large value of γ (remember that γ has to be close to one for the fulfillment of Zipf’s law).

Remarkably, Figure 1(right) also shows that all the marginal distributions present a characteristic, roughly the same shape, with the only difference being on the scale parameter of the frequency distribution, which is determined by the mean frequency $\langle \rho_{kk^{\prime }}\rangle$ (denoted generically in the figure as $\langle \rho_{emp}\rangle$). This means, as shown in the figure, that the distribution $g(\rho_{emp})$, when multiplied (rescaled) by $\langle \rho_{emp}\rangle$, can be considered, approximately, as a function that only depends of the rescaled frequency, $\rho_{emp}/\langle \rho_{emp}\rangle$, independently on which potential $\rho_{kk^{\prime }}$ one is considering. In terms on the distribution of ranks this scaling property translates into the fact that $\rho_{emp}/\langle \rho_{emp}\rangle$ can be considered a function of only r/V.

For the fitting we have used the method proposed in [66,67], based on maximum-likelihood estimation and Kolmogorov-Smirnov goodness-of-fit testing. This method lacks the problems presented in the popular Clauset et al.’s recipe [3,68,69]. The fitting method is applied to ρ as a random variable (instead than applied to *r* [16]); this choice presents several important advantages, as discussed in [65]. The outcome of the method is a estimated value of the exponent γ together with a value of ρ, denoted by *a*, from which the power-law fit, Equations (Equation 12) and (Equation 13), is non-rejectable (with a *p*-value larger than 0.20, by prescription). Although other distributions different than the power law can be fitted to the marginal data (e.g., lognormal [67]) our purpose is not to find the best fitting distribution, but just to evaluate how much Zipf’s power law depends on a possible Zipf’s behavior of the marginals.

### 3.3. Word Distributions

Figure 2 shows that the optimization succeeds in getting values of the theoretical marginal distributions very close to the empirical ones. However, despite the fact the target of the optimization are the marginal distributions (whose empirical values are the input of the procedure), we are interested in the distribution of words, whose empirical value is known but does not enter into the procedure, as this is the quantity we seek to “explain”. Zipf’s rank-frequency representation allows us to display in one dimension the six-dimensional nature (from our point of view) of the word frequencies; for the empirical word frequencies this is shown in Figure 3. We find that the distribution is better fitted in terms of an upper truncated power law [66,71], given, as in Equation (Equation 13), by $g\left(\rho \right)\propto 1/\rho^{\gamma +1}$ but in a finite range a≤ρ≤b<∞ (the untruncated case would be recovered by taking b→∞). This corresponds, in the continuum case, to a cumulative distribution $G\left(\rho \right)\propto 1/\rho^{\gamma }-1/b^{\gamma }$, and to a rank-frequency relation
$$\rho \propto \frac{1}{{\left(r+V/b^{\gamma }\right)}^{1/\gamma }},$$
which coincides in its mathematical expression with the so-called Zipf-Mandelbrot distribution (although the continuous fit makes *r* a continuous variable; remember that *V* is the number of types). The fitting procedure is essentially the same as the one for the untruncated power law outlined in the previous subsection, with the maximization of the likelihood a bit more involved [66,67].

In Figure 3 we also display the theoretical result, $P^{II}$, Equation (Equation 8), arising from the solution of Equation (Equation 10). We see that, qualitatively, $P^{II}$ has a shape rather similar to the empirical one. Both distributions fulfill Zipf’s law, with exponents γ equal to 0.89 and 0.81, respectively. We also see in the figure that the quantitative agreement in the values of the probability ($P^{II}$ and $\rho_{word}$) is rather good for the smallest values of the rank (r<10); however, both curves start to slightly depart from each other for r>10. In addition, the rank values are associated with the same word types for r≤6 (the, of, and, to, a, in), but for larger ranks the correspondence may be different (r=7 corresponds to i in one case and to that in the other). If we could represent $\rho_{word}$ and $P^{II}$ in six dimensions (instead that as a function of the rank) we would see more clearly the differences between both.

Zipf’s law is, in part, the reason of this problem, as for r≥10 the difference in probabilities for consecutive ranks becomes smaller than 10 %, see Equation (Equation 11), and for r≥100 the difference decreases to less than 1 % (assuming γ≃1). Therefore, finite resolution in the calculation of $P^{II}$ will lead to the “mixing of the ranks.” However, the main part of the problem comes from the unability of the algorithm in some cases to yield values of $P^{II}$ close to the empirical value, $\rho_{word}$, as it can be seen in the scatter plot of Figure 4 (in agreement with [55]). The entropy of the theoretical word probabilities turns out to be S=9.90 bits, somewhat larger than the corresponding empirical value 8.35 bits. If we truncate this distribution, eliminating probabilities below 10,000/1,597,358,419 $≃6\times 10^{-6}$ (as in the empirical distribution) we get S=8.88 bits, still larger than the empirical value, which simply means that real language has more restrictions than those imposed by the model. Existing (empirical) words for which the algorithm yields the lowest theoretical probabilities are enumerated in the caption of the figure. Curiously, as it can be seen, these are not particularly strange words.

An interesting issue is that the maximum-entropy solution, Equation (Equation 8), leads to the “discovery” of new words, or, more properly, pseudowords. Indeed, whereas the empirical corpus has V=5081 (number of word types), the theoretical solution leads to V= 2,174,013 (words plus pseudowords). Most of these pseudowords have very small probabilities; however, there are others far from being rare (theoretically). In this way, the most common pseudoword (theoretical word not present in the empirical corpus) is whe, with a theoretical rank r=40 (it should be the 40-th most common word in English, for length six or below, following the maximum-entropy criterion). Table 2 provides the first 25 of these pseudowords, ranked by their theoretical probability $P^{II}$. We see that the orthography of these pseudowords looks very “reasonable” (they look like true English words). On the other side, the most rare pseudowords, with probability $P^{II}∼10^{-30}$, are nearly impossible English words, as: sntnut, ouoeil, oeoeil, sntnu, snsnua… (not in the table).

### 3.4. Values of Lagrange Multipliers and Potentials

We have established that, for a given word, the value of its occurrence probability $P^{II}$ comes from the exponentiation of the sum the 15 interaction potentials between the six letter positions that constitute the word (in our maximum-entropy approach). Therefore, the values of the potentials (or the values of the Lagrange multipliers) determine the value of the probability $P^{II}$. It is interesting to investigate, given a potential or a multiplier (for instance $\lambda_{12}$), how the different values it takes ($\lambda_{12}\left(aa\right),\lambda_{12}\left(ab\right)$, etc.) are distributed. Curiously, we find that the 15 different potentials are (more or less) equally distributed, i.e., follow the same skewed and spiky distribution, as shown in Figure 5(left).

One can try to use this fact to shed some light on the origin of Zipf’s law. Indeed, exponentiation is a mechanism of power-law generation [44,68]. We may arguee that the sum of 15 random numbers drawn from the same spiky distribution has to approach, by the central limit theorem, a normal distribution, and therefore, the exponentiation of the sum would yield a lognormal distribution for $P^{II}$ (i.e., a lognormal shape for $g(P^{II})$). However, this may be true for the central part of the distribution, but not for its rightmost extreme values, which is the part of the distribution we are more interested in (high values of $P^{II}$, i.e., the most common words). Note also that, in practice, for calculating the probability of a word, we are not summing 15 equally distributed independent random numbers, as not all the words are possible; i.e., there are potentials that take a value equal to infinite, due to forbidden combinations, and these infinite values are not taken into account in the distribution of the potentials. An additional problem with this approach is that, although most values of the potentials converge to a fix value (and the distribution of potentials shown in the figure is stable), there are single values that do not converge, related to words with very low probability. These issues need to be further investigated in future research. In addition, Figure 5(right) shows, as a scatter plot, the dependence between the value of each potential and the corresponding two-letter marginal probability. Although Equation (Equation 10) seems to indicate a rough proportionality between both, the figure shows that such proportionality does not hold (naturally, the rest of terms in the equation play their role).

## 4. Discussion

We have generalized a previous study of Stephens and Bialek [55]. Instead of restricting our study to four-letter words, we consider words of any length from one to six, which leads to greater computational difficulties, and employ a much larger English corpus as well. We perform an analysis of the fulfillment of Zipf’s law using state-of-art statistical tools. Our more general results are nevertheless in the line of those of [55]. We see how the frequency of occurrence of pairs of letters in words (the pairwise marginal distributions), together with the maximum-entropy principle (which provides the distribution with the maximum possible randomness), constrain the probabilities of word occurrences in English.

Regarding the shape of the distributions, the agreement between the maximum-entropy solution for the word distribution and its empirical counterpart is very good at the qualitative level, and reasonably good at the quantitative level for the most common words, as shown in Figure 3. Moreover, new possible English words, or pseudowords, not present in the corpus (or, more exactly, in the subcorpus we have extracted) have been “discovered”, with hypothetical (theoretical) values of the occurrence probability that vary along many orders of magnitude. However, regarding the probabilities of concrete words, the method yields considerable scatter of the theoretical probabilities (in comparison with the known empirical probabilities), except for the most common words, see Figure 4.

As two by-products, we have found that the pairwise (two-letter) occurrence distributions are all characterized by a well defined shape, see Figure 1(right), and that the distributions of the 15 different interaction potentials are nearly the same, see Figure 5(left). The latter is an intriguing result for two reasons. First, the fact that the values that the 15 interaction potentials take are more or less the same for all of them (Figure 5(left)) seems to indicate that all the potentials are equally important; nevertheless, remember the potentials are undefined with respect one additive constant, and comparison of their absolute values is misleading. Second, not only the values that the potentials take are nearly the same, but they seem to be equally distributed. We have tried to relate, without success yet, this distribution to other skewed and spiky distributions that appear in complex and correlated systems, such as the so-called Bramwell-Holdsworth-Pinton (BHP) distribution [72], the Tracy-Widom distribution, or the Kolmogorov-Smirnov distribution [73,74].

Despite our results, one could still abandon the all-to-all interaction and embrace instead nearest-neighbor coupling (this may seem similar to a Markov model, however, the study of the possible similarities or not should be the subject of a future work). Nearest-neighbor coupling reduces the number of potentials from 15 to 5 (with open boundary conditions), with the subsequent computational simplification. A further reduction would be to impose that all potentials are the same (i.e., they do not depend on letter positions, only on difference of positions, e.g., $\lambda_{12}=\lambda_{23}$, etc.). This leads to only one potential (in the case of nearest-neighbor interaction; 5 potentials in the all-to-all case). It would be interesting to compare these modifications with the original model and to confirm that they lead to much worse results; this is left for future research.

An extension towards a different direction would be to use phonemes or syllables instead of letters as the constituents of words. We urge the authors of the corpus in [63] to provide the decomposition of the words in the corpus into these parts. Naturally, other languages than English should be studied as well. An interesting issue is if our approach (which is that of [55]) can shed light on other linguistic laws; in particular, the word-length law [75,76] and the brevity law (also called Zipf’s law of abbreviation [75,76,77]). Therefore, we could verify up to which point longer words have smaller theoretical probabilities, and if the robust patterns found in [76] are also valid for the maximum-entropy solution. Furthermore, we could quantify the role of the pairwise interaction potentials in determining the length of the word (the “brevity”), looking at the interaction of any of the 26 letters with the blank. Alternatively, one could study the word frequency distributions at fixed length [76], and check if the potentials are stable for different word lengths. Finally, let us mention that the approach presented here has also been applied to music [78]. This, together with the applications in neuronal, biochemical, and genetic networks mentioned above ([55] and references therein) confirms the high interdisciplinarity of this approach.

## Figures and Tables

**Figure 1 entropy-22-00179-f001:**
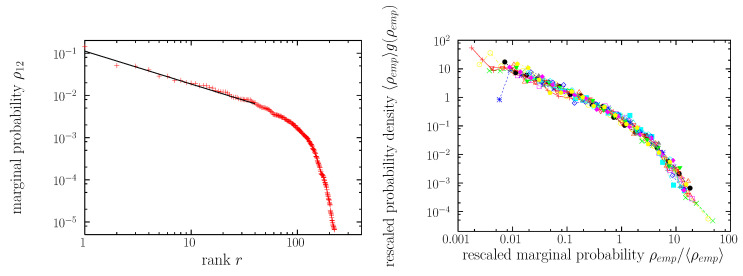
Empirical two-letter marginal distributions (for word length not larger than 6 letters). **Left**: The distribution $\rho_{12}$ is represented in terms of the rank-frequency plot [corresponding to Equation (Equation 11)]. The most common values of $\rho_{12}$ correspond to the following pairs: th, an, of, to, he, in, a□, ha, wh, wa, … The power-law fit from Table 1 is shown as a straight line, with exponent 1/γ=0.78. **Right**: All 15 two-letter marginals are represented in terms of the distributions of the value of the marginal probabilities, $\rho_{12},\rho_{13},\ldots \rho_{56}$ (denoted in general as $\rho_{emp}$). All the distributions have been shifted (in log-scale) by rescaling by their mean values $\langle \rho_{emp}\rangle$, see [70]. This makes apparent the similarities between all the two-letter marginal distributions, except for a scale factor given by $\langle \rho_{emp}\rangle$. Values below the mean ($\rho_{emp}<\langle \rho_{emp}\rangle$) can be fitted by a truncated power law, with exponent $1+\gamma^{\prime }≃0.9$ (not reported in the tables). The tail (large $\rho_{emp}$) is well fitted by power laws, with the values of exponents 1+γ in Table 1.

**Figure 2 entropy-22-00179-f002:**
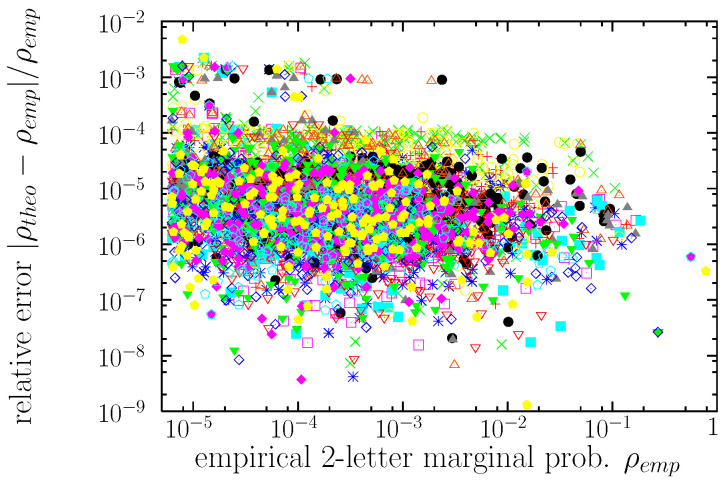
Comparison between the empirical two-letter marginal distributions $\rho_{emp}$ and the theoretical ones $\rho_{theo}$ obtained from the improved iterative-scaling optimization procedure [61,62]. The relative error between both values of the marginal probability is shown as a function of the empirical value, for the 15 marginals.

**Figure 3 entropy-22-00179-f003:**
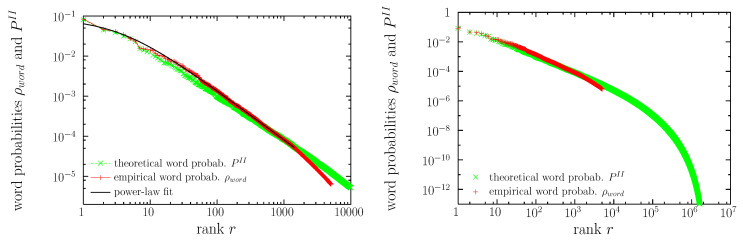
Empirical ($\rho_{word}$) and maximum-entropy theoretical ($P^{II}$) word occurrence probabilities in the rank-frequency representation, together with the power-law fit of the distribution of frequencies for the empirical case. The same distributions are shown at two different scales. **Left**: only ranks below 10,000. **Right**: only probabilities (frequencies) above $10^{-13}$.

**Figure 4 entropy-22-00179-f004:**
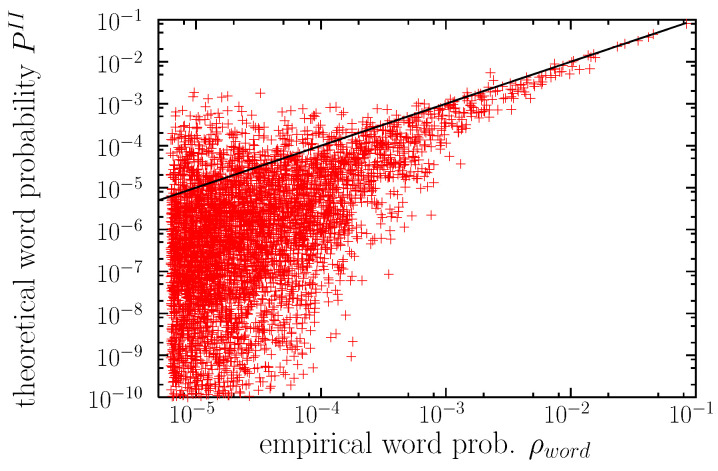
Maximum-entropy theoretical probability $P^{II}$ for each word type in the sub-corpus as a function of its empirical probability (relative frequency) $\rho_{word}$. The straight line would signal a perfect correspondence between $P^{II}$ and $\rho_{word}$. Values of $P^{II}$ below $10^{-10}$ are not shown. Words with the lowest $P^{II}$ (in the range $10^{-17}$–$10^{-15}$) are shaggy, isaiah, leslie, feudal, caesar, yankee, opium, yields, phoebe, sydney.

**Figure 5 entropy-22-00179-f005:**
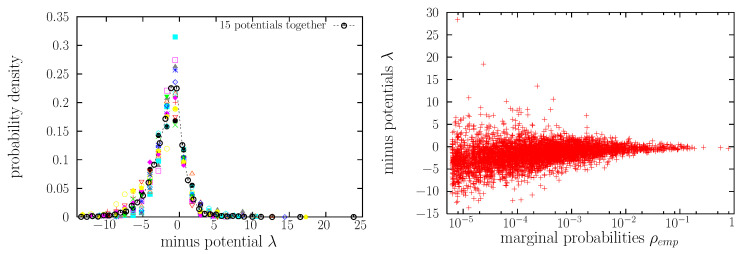
**Left**: Empirical probability densities of the 15 individual potentials (with a negative sign) and the probability density of the 15 aggregated data sets. **Right**: Value of the Lagrange multiplier (which corresponds to the interaction potential with a negative sign) for each pair of letters (and positions) as a function of the corresponding marginal probability.

**Table 1 entropy-22-00179-t001:** Results of power-law fitting of the form $g\left(\rho \right)\propto 1/\rho^{\gamma +1}$ (for a≤ρ≤b) applied to the 15 empirical two-letter marginal distributions (with b=∞), to the empirical word frequency $\rho_{word}$ and to the theoretical maximum-entropy solution $P^{II}$. The empirical distribution for words of any length, $\rho_{all\;word}$, is also shown, in order to compare it with $\rho_{word}$. *V* is the number of types (pairs of letters or words); $\rho_{max}$ is the highest empirical frequency; $j_{max}$ is the corresponding type (pair of letters or word type; the next highest-frequency types appear in brackets); o.m. is the number of orders of magnitude in the fit, $\log_{10}\left(\rho_{max}/a\right)$; *v* is the number of types that enter into the power-law fit; σ is the standard error of the fitted exponent; and *p* is the p–value of the goodness-of-fit test. The ratio v/V ranges from 0.09 to 0.3. Only words of length from 1 to 6 are taken into account. Blanks are not considered in the marginals. 50 values of *a* and *b* (when *b* is not fixed to ∞) are analyzed per order of magnitude, equally spaced in logarithmic scale. p–values are computed from 1000 Monte Carlo simulations. Fits are considered non-rejectable if p≥0.20.

Distribution	*V*	$\rho_{\max}$	$j_{\max}$	*a* $\left(\times 10^{-4}\right)$	*b*	o.m.	*v*	γ±σ	*p*
$\rho_{12}$	223	0.143	th (an, of, to, he)	63.1	*∞*	1.36	40	1.282 ± 0.213	0.21
$\rho_{13}$	471	0.146	te (i□, o□, a□, ad)	16.6	*∞*	1.94	133	1.138 ± 0.097	0.24
$\rho_{14}$	455	0.038	t□ (a□, o□, i□, h□)	34.7	*∞*	1.04	81	1.391 ± 0.156	0.23
$\rho_{15}$	391	0.043	t□ (a□, o□, i□, h□)	36.3	*∞*	1.07	78	1.433 ± 0.175	0.28
$\rho_{16}$	285	0.042	t□ (a□, o□, w□, i□)	69.2	*∞*	0.78	45	2.110 ± 0.324	0.23
$\rho_{23}$	309	0.160	he (f□, o□, □□, nd)	57.5	*∞*	1.44	42	1.207 ± 0.197	0.29
$\rho_{24}$	361	0.049	h□ (n□, o□, f□, e□)	60.3	*∞*	0.91	50	1.466 ± 0.210	0.24
$\rho_{25}$	334	0.057	h□ (o□, n□, e□, a□)	52.5	*∞*	1.04	53	1.309 ± 0.183	0.29
$\rho_{26}$	240	0.055	h□ (o□, n□, e□, a□)	145.0	*∞*	0.58	21	2.576 ± 0.627	0.22
$\rho_{34}$	330	0.048	□□ (e□, d□, s□, t□)	83.2	*∞*	0.76	36	1.764 ± 0.340	0.41
$\rho_{35}$	371	0.039	□□ (e□, d□, s□, r□)	50.1	*∞*	0.89	57	1.359 ± 0.190	0.28
$\rho_{36}$	273	0.045	□□ (e□, d□, r□, t□)	75.9	*∞*	0.78	44	1.935 ± 0.298	0.32
$\rho_{45}$	278	0.051	□□ (e□, t□, n□, h□)	87.1	*∞*	0.77	35	1.579 ± 0.270	0.33
$\rho_{46}$	244	0.044	□□ (e□, t□, n□, l□)	100.0	*∞*	0.64	31	1.946 ± 0.378	0.28
$\rho_{56}$	154	0.115	□□ (e□, s□, d□, t□)	72.4	*∞*	1.20	34	1.140 ± 0.201	0.58
$\rho_{all\;word}$	11042	0.071	the (of, and, to, a)	1.0	0.073	2.85	925	0.925 ± 0.030	0.25
$\rho_{word}$	5081	0.084	the (of, and, to, a)	0.5	0.087	3.20	1426	0.811 ± 0.023	0.31
$P^{II}$	2174013	0.081	the (of, and, to, a)	0.2	0.083	3.53	2947	0.886 ± 0.017	0.38

**Table 2 entropy-22-00179-t002:** Most common theoretical words from the maximum-entropy procedure that are not present in the analyzed sub-corpus. In fact, all of these theoretical words are present in the original complete corpus (but not in our sub-corpus as we have disregarded frequencies smaller than 10,000). We can distinguish four different cases: the (theoretical) word does not exist in a dictionary (∄) and can be considered a pseudoword (and appears in the complete corpus probably as a misspelling); the word exists in a dictionary as a word (∃); the word appears in a dictionary as an archaism (arch.); and the word appears as an abbreviation (abbrev.). *r* is (theoretical) rank and $P^{II}$ is (theoretical) probability.

*r*	$P^{\mathrm{II}}$	Word	Case
40	2.88 $\times \;10^{-3}$	whe	∄
48	2.20 $\times \;10^{-3}$	wis	abbrev.
52	1.95 $\times \;10^{-3}$	mo	abbrev.
61	1.74 $\times \;10^{-3}$	wast	arch.
64	1.69 $\times \;10^{-3}$	ond	∄
71	1.52 $\times \;10^{-3}$	ar	abbrev.
77	1.40 $\times \;10^{-3}$	ane	∃
87	1.24 $\times \;10^{-3}$	ald	abbrev.
89	1.21 $\times \;10^{-3}$	bo	∃
92	1.16 $\times \;10^{-3}$	thes	∄
94	1.10 $\times \;10^{-3}$	hime	∄
98	9.83 $\times \;10^{-4}$	hive	∃
102	9.45 $\times \;10^{-4}$	thise	∄
103	9.39 $\times \;10^{-4}$	af	abbrev.
110	8.80 $\times \;10^{-4}$	wer	∄
117	8.16 $\times \;10^{-4}$	thay	∄
118	8.16 $\times \;10^{-4}$	hes	∄
123	7.88 $\times \;10^{-4}$	wath	∃
125	7.82 $\times \;10^{-4}$	hor	abbrev.
127	7.60 $\times \;10^{-4}$	sime	∄
134	7.22 $\times \;10^{-4}$	tome	∃
135	7.21 $\times \;10^{-4}$	har	∃
141	6.94 $\times \;10^{-4}$	thit	∄
143	6.86 $\times \;10^{-4}$	mas	abbrev.
146	6.77 $\times \;10^{-4}$	hew	∃

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
