# Peer review of "From Boltzmann to Zipf through Shannon and Jaynes"

_entropy, 2020, doi:10.3390/e22020179_

Round 1
Reviewer 1 Report
REVIEW: From Boltzmann to Zipf through Shannon and Jaynes
In this article, the authors review and extend a maximum-entropy formalism and its application to pairwise interaction of letters in words, using a model based on the interpretation of word frequency as arising from the interaction potential between its letters (Stephens & Bialek). The zipfian word distribution of English texts (Standardized Project Gutenberg Corpus) is discussed, as well as the adequation of Shannon entropy, so authors find that the model of Stephens & Bialek is able to recover Zipf’s law for words in texts but with considerable scatter of the values of the probabilities, extending and improving Stephens & Bialek’s results.
After the review, I recommended the manuscript for publication only when the main issues exposed are improved or corrected.
Main issues
1.- The work has a multidisciplinary interest that matches Entropy objectives and the Special Issue to which it has been sent but, as a general comment to improve the article, there should be a greater effort to show this interdisciplinarity, in this case to approach the paper to potential readers of the field of mathematical Linguistics or even Psychology, Engineering (Information Theory) and Statistical Learning.
2.- Title. The title is very suggestive and seems to advance that the article will have a more humanistic approach. Unfortunately this is not the case. I recommend either changing the title or, better yet, maintaining it but “explaining the title” more transversely and clearly in both the introduction and the discussion, i.e. closing the article.
3.- Abstract. In my opinion, the 'effort' in explaining the title made in the abstract should be left for the introduction. On the other hand, it would suffice to cite the year of the publication of Stephens & Bialek (2010) and more clearly explain that the article is an extension of this work, for English, increasing both the size of the corpus and the number of letters of the model. Specifically I would suppress(move to introduction): “Indeed, Jaynes’ maximum-entropy principle, with the constrains given by every empirical two-letter marginal distribution, leads to a Boltzmann distribution for word probabilities, with an energy-like function given by the sum of all pairwise (two-letter) potentials. The improved iterative-scaling algorithm allows us finding the potentials from the empirical two-letter marginals.” And continue: “Appling Stephens and Bialek’s formalism…” explaining then the main contributions of the paper.
4.- Keywords: They should be added. It is a serious oversight that prevents publication in Entropy.
5.- 1. Introduction. In this section, although not important, the authors surprisingly begin the introduction to Zipf's law in reverse: they first expose its ubiquity and then end up commenting on their "particular interest" in Linguistics.
6.- Although called "Zipf's law" the authors forget to cite other precursors prior to Zipf (Estoup, Pareto or Condon) and Zipf's own works (!), focusing on references by more modern authors (some maybe unnecessary). At least in the first paragraph, Zipf's works should be duly cited, and if they really want to be consistent with the title, also link the connection between Boltzmann's works and Shannon and Jaynes' approach (i.e. rewriting here the succinct effort done in the abstract).
7.- On the other hand, since they are going to focus on the study of Zipf's law in linguistic corpus, It would not be bad to include some brief paragraph about the variability of Zipf's law in these linguistic corpus (according to the language of study, the size of the corpus, n-grams, pathology, language acquisition…), in speech corpus or transcripts at prephonemic levels, etc., which at the end allows a richer discussion about future work (i.e. see https://journals.plos.org/plosone/article?id=10.1371/journal.pone.0053227 ).
8.- Lines 59-60. Authors claim that: “The framework is that of word occurrence in texts, and words are considered as composed by letters that interact in pairs.” Although they give the reference it would not be bad if they briefly exposed what this interaction in pairs consists of (with the following letter? With the previous one?, with both in the chain?).
9.- 2. Maximum entropy and pairwise interactions. I especially like this section, because there is a theoretical background that allows us to connect information theory, complexity and corpus linguistics. My suggestions here go in the line of really making this approximation. First, after the exposure of the basic theoretical framework, on line 86 the authors strongly affirm (I don't know if following the jokes of the footnote): “Needless to say, we have no idea yet what the energy Ej of a word is.” One thing is that the units in which the energy or acoustic power is measured are relative and the other is that the authors refer here "to the energy of a written word". The bases of bioacoustics are well known not only in human languages but in animal communication (see Fletcher’s work in https://www.springer.com/gp/book/9781493907540 or the whole book...), and the authors should be documented in this regard or, if it is not their objective, delete the sentence. However, I suggest to explore this topic a bit, because it is often forgotten that writing is the reflection of orality, and even the analogue of a Gutenberg-Richter law in different levels of speech has been recently studied (https://royalsocietypublishing.org/doi/10.1098/rsif.2014.1344 and https://www.nature.com/articles/srep43862 ).
10.- Please provide a general reference to introduce equation (2) (before of after equation 2, i.e. line 85).
11.- I would recommend changing 'feature function' to 'letter function'. Because, at what moment is another 'feature' of the word used, other than the letter it contains? I would understand that they kept 'feature' if for example other features were explored (semantic, syntactic…), but it is not the case.
12.- In the equation between 3 and 4 (not all the indented are numbered, by the way) authors, referring to the Lagrange multipliers, “inverted their sign with respect the previous examples, for convenience”. Although it is obvious, I would clarify it before giving the equation, and referring to the fact that the sign change is made with respect to equation 1.
13.- Between lines 94-96, do the authors recognize that it is a kind of monkey typing model? This is a key issue. Are the probabilities of each letter obtained by 'copying' the probabilities of each letter in the corpus, with a kind of Markovian dependence? What do the authors suggest about a Markovian approach?
If not here, in the discussion they should extend comments about this, that is, about the relationship –or not- of their work with the approximations of random text generations and Markov models ( i.e. https://arxiv.org/pdf/1207.1872.pdf https://iopscience.iop.org/article/10.1088/1742-6596/936/1/012028 and more generally ref[18] http://www.eecs.harvard.edu/~michaelm/CS222/powerlaw.pdf ) .
14.- 3.Data and results. After line 108, the authors say that they have a usual “computational limitation”, which finally (lines 124-125) leads them to assume that they have skewed their sample (to the subset of words with length smaller or equal to 6). Therefore, in a large corpus low-frequency words, statistically longer according to the law of brevity, have surely been eliminated, affecting the Zipf exponent, as they recognize, and other issues (i.e. review https://pdfs.semanticscholar.org/33ae/2217922ded2a7af200e13560c59af9801e7c.pdf ) . How the exponent is affected, and how do authors intuit their results would be if the hapax legomena and dis legomena (Large Number of Rare Events, following Baayen) were –or not- eliminated? If not here please answer this question in Discussion.
15.- Figure 1. The approximate exponent (or exponents) of the distributions should be added in both figures.
16.- Table 1. It would be interesting and illustrative to readers to include a column, after the column 'distribution', with some “Examples” or “Common values”, with the most common values of rhos, as has been done with the listing in figure 1, for rho12 (th, an, of, to, he, in, a_, ha, wh, wa,…)
17.- In the final paragraph of section 3.3. (lines 187-196), the authors are confused when talking about 'new words'. There is a very well-known concept in psychology and linguistics: 'pseudoword'. This matches the intuition of the authors (“…the ortography of these words looks very “reasonable” (they look like true English words)”), so I strongly recommend authors to rewrite this paragraph, discussion and the caption of table 2, from the perspective of the 'generation of pseudowords', which is what the authors have done partially (They also generate ‘impossible words’…).
18.- This topic remember to me Shannon’s classical work (http://www.math.harvard.edu/~ctm/home/text/others/shannon/entropy/entropy.pdf ) page 4, figure 2 about telegraphy: a similar figure about word generation could be very interesting. Is this artificial language generated a ‘second order approximation’ (Shannon, last paragraph p.6-7)? Following Shannon, what are the probability tables of each pair of letters in this model and in the database (=excellent supplementary material)? Please explain this issue better.
19.- On the other hand, continuing with the linguistic confusions, Table 2 includes real English words (i.e. hive), some words in disuse but existing in old English (accepted at the present moment in Dictionaries, i.e. wast) and pseudowords… One option would be to add a column specifying whether it is real words, pseudowords or impossible words (which have been removed from this sample in table 2, without a clear reason…). Although physicists may not care much about that, they should take special care of these details to reach a wider audience (Entropy's goal).
20.- In section 3.4., the effort of the authors to connect their study with Linguistics is really missed. The authors acknowledge that the method fails when faced with the uncommon words (what is to be expected given the truncation in the length of the words up to the sixth character). Can any correlation be established with this results and linguistic law of brevity or with linguistic optimization models based on information theory? (i.e. https://arxiv.org/abs/1906.01545)
21.- Discussion. Finally, I think the study is powerful enough to expect the authors to be somewhat more daring in the discussion. I mean, it is important to go beyond mere (important) calculations, and consider how their results can change some of Shannon's original conclusions or compare to other related work (i.e. Shannon’s classical work, entropy rate and word estimations via cognitive experiments, Amazon Mechanical Turk,…). There are an enormous variability between English and other languages, so to what extent this approach depends on the number of characters considered? It is an aspect to mention, at least in the conclusions, and to propose future work in other languages (comparative studies) in this direction.
If not all of them, I think the authors should try to speculate a bit in the discussion about the answers (or future lines of research) at least of some of my questions, and, as I said at the very beginning, end the article in a way more in line with the very suggestive title proposed.
Minor issues
1.- Line 30: “…mention monkey typing, also called intermittent silence [21,22],” I suggest more properly: “mention monkey typing, also called intermittent silence [21] refuted in [22],…”
2.- I recommend to change ref [26] for this one more specific of the same authors: https://iopscience.iop.org/article/10.1088/1367-2630/18/9/093010#njpaa3365bib17
3.- Lines 50-52. Authors claim that: “The main idea is that the probability is exponential with the energy, but the energy is logarithmic with size, resulting in an overall power law for sizes.” Reference of Peterson et al. should be cited at the end of this sentence.
4.- Typo after line 84 (no number line): “contrains” should be “constrains”
5.- Please include Zipf’s references after line 129 “…representation (also used by Zipf) considers…”
Reviewer 2 Report
This manuscripts investigates the origin and significance of Zipf's law using maximum entropy arguments. It contains a remarkably complete and extremely useful introduction to the problem, pointing to the diversity of existing works related to Zipf's law. I believe it is a strength of this manuscript to acknowledge the different approaches and to go deeper into one of the previous directions (the one proposed by Stephens and Bialek), recognizing that there is no unique answer to the question of the origin of Zipf's law. The manuscript is clearly written and the results are supported by the analysis. I recommend this manuscript for publication after the authors consider the comments and suggestions below.
Comments:
1) Up to line 94 I was interpreting the null model as a Markov process of order 1 or 2 (in letters), as the one considered by Shannon in his famous paper. It'd be interesting to clarify more clearly at this stage the differences to this process, both in terms of its definition (e.g., in the model considered in the manuscript \rho_1(c) is different from \rho_2(c)) and in terms of consequences (e.g., what is the rank-frequency distribution of words and bi-grams for the Markov process at the letter level?).
2) The restriction of word types to those with a frequency of 10,000 is very severe, it excludes mode of the word types and 4 orders of magnitude in frequency. This is against the spirit of Zipf's law, which is precisely interested in describing the tails and multiple scale of words. What is the justification for this choice? What is the effect on the results?
3) At the end of page 6 the authors mentioned that the two-letter marginals is bivariate and oppose this to the univariate rank distribution (similarly, later it is mentioned that the distributions are 6 dimensional). I found this confusing because it suggests that a projection is being done. In fact, these distributions are defined over the space of symbols V, so that the support of \rho_12 is in fact a set of size V^2, or (V+1)^2 if one includes the blank space. This is the same size used in the rank distribution.
4) Table 1. I think the results reported here are problematic, in particular the computation and interpretation of the p-value:
The authors say that the fits are "non-rejectable", what does that mean? My understanding is that the fits should be rejectable, and that the finding is that these particular ones were not rejected. Please check if this is what it is meant by this term. The choice of threshold for rejection, p=0.2, is very unusual (typically it is either 0.01, 0.05, or 0.1) and it'd be important to justify why such a large value is chosen in this case. It gives the impression that the choice was made after seeing the results, which is what one should absolutely not do. The authors report p-values for 18 cases in the table. If the data is sampled from the distribution being tested, we should expect a flat distribution of p-values and, at a threshold of p=0.2, that on average 0.2*18 = 3.6 cases to be rejected. Instead, no case was rejected and in fact all p-values are in the range between 0.21 and 0.58 (most of them smaller than 0.3). This is a strong indication that there is something wrong with the way these p-values are being computed, or further clarifications are needed. The range in which the fit is being performed is relatively small, less than 2 decades. One possibility to explain the anomalies of the p-values is that this method is choosing a fitting range for which results have specific p-values. This is of course not a good way to test hypothesis.5) Fig. 4. It'd be interesting to see predictions for the fluctuations around the expected value of the model, e.g. due to finite sampling. In this way we could compare whether the observed fluctuations are compatible with the model, getting some insights on aspects the model get right and wrong.
6) line 248, and conclusion. I found it hard to understand and interpret the finding that all potentials are equally important regardless of the distance between the letters. First, I can not see how this conclusions can be drawn from Fig. 5. Given its importance, it'd be nice to see a specific panel on the dependence of the length. More importantly, it left me thinking whether this is not against the whole motivation of this approach of considering pairwise interactions between letters. The authors mention the possibility of extending this to the level of phonemes, which would possibly have a more relevant lingusitic interpretation, would that be a way of justifying this approach? If so, how can it be that fair away phonemes interact as much as phonemes next to each other? This seems against intuition. Alternatively, why should we stop at pairwise interactions and not include interaction between 3 letters or phonemes? I think I'm missing a better intuition or motivation for this particular model (possibly in the introduction?) that would allow me to understand the paper as a whole and the findings obtained here.
Minor points and typos:
line 8, it is not clear what "The" refers to. I guess the authors want to say "We propose an ..."
line 9: "Appling" -> "Applying"
line 85: "term" -> "notation" (there is no term "can" in the equation)
line 106: "ride" -> "rid" and "symmetries" -> "symmetry"
line 126: the authors refer to "4-order of magnitude" a line after saying that the theoretical maximum is 676 (much less than 10000 or four orders). I guess the authors are thinking about the right panel of the figure, after re-scaling and combining all cases. Maybe the text could be reformulated to be more explicit.
line 129+1: It is not clear to me what "approximated" means here. Isn't this just an alternative representation?
line 155: "instead than to" -> "instead of"
line 181: the larger entropy results are interesting, yet not surprising. They simply mean that actual language is more restricted than this simple model.
table 2: this is an interesting discussion. The authors could also highlight that the word "hive" is a good English word that just happened not to be inside their threshold but that is then "predicted" by the model.
Appendix: it'd be nice if the authors could publish also the codes they used to perform the analysis.
